# Repeat low order caesarean delivery, risk factors for complications: A retrospective, longitudinal study

Orna Reichman*[☉], Misgav Rottenstreich[☉], Hen Y. Sela, Rachel Michaelson-Cohen, Zvi Ehrlich, Reut Rotem, Sorina Grisaru-Granovsky

Faculty of Medicine, Department of Obstetrics and Gynecology, Shaare Zedek Medical Center, Hebrew University of Jerusalem, Jerusalem, Israel

☉ These authors contributed equally to this work.
* orna.reich@gmail.com

**Data Availability Statement:** All relevant data are within the manuscript and its Supporting Information files.

## Abstract

One-third of cesarean deliveries (CDs) are repeat operations, of which the majority are low-order, second (CD2) and third (CD3). The study objectives were to identify risk factors for a complicated maternal CD among women undergoing a repeat low-order CD and to develop a predictive model for at-risk women. A retrospective longitudinal follow-up study was conducted in a single medical center, during 2005–2016. Women who underwent both CD2 and CD3 at the site were included. Those with placenta accreta or a caesarean hysterectomy were excluded. A composite complicated maternal CD was defined by either uterine rupture/ dehiscence, blood transfusion, relaparotomy, admission to the intensive care unit or prolonged operative time >90th percentile. Data was analyzed comparing between CD2 to CD3, each woman served as her own control. Univariate analysis followed by a multivariate logistic regression modeling were performed with an OR of 95% CI defining significance. The study group comprised of 1,331 women. A complicated CD occurred in 159 (12%) vs. 226 (17%) of CD2 vs. CD3 respectively, (p<0.001). Women with a complicated CD2 were at higher risk for complications in CD3, aOR 2.3 (95% CI 1.5, 3.3). Sub-Saharan African origin and preterm delivery at CD3 were both risk factors for a complicated CD3, aOR 3.7 (95% CI 1.9, 7.3) and aOR 1.7 (95% CI 1.1, 2.7), respectively. The multivariate regression model included 1328 cases, was statistically significant, $\chi2(7) = 50.760$, p <0.001, explained 6.3% of the variance of composite complicated maternal CD3 and correctly classified 82.9% of cases. Although a complicated CD2, Sub-Saharan African origin and preterm delivery are risk factors for maternal complications in CD3, it is hard to predict which specific women will experience complications. Sensitivity, specificity, positive and negative predictive value of a complicated CD2 for detecting complications in CD3 were 21%, 90%, 30% and 85% respectively.

## Introduction

In the past few decades there has been a sharp increase in the rate of caesarean deliveries (CDs) worldwide, with a global average of 21% [1]. This trend is most noticeable in middle to

**Funding:** The author(s) received no specific funding for this work.

**Competing interests:** The authors have declared that no competing interests exist.

high income countries, reaching up to 50% of all deliveries [1–3]. Approximately one-third of CDs are repeat surgeries [4].

Large observational studies have repetitively shown that high order repeat CDs (≥4 CD), as well as placenta accreta spectrum, are associated with maternal and fetal complications [5–8]. These include severe maternal hemorrhage, need for blood transfusions, damage to adjacent organs, thromboembolic events, uterine rupture, peripartum hysterectomy, and neonatal morbidity and mortality [5–8]. However, only 8% of repeat CDs are high order CDs and even less are complicated by the placenta accreta spectrum (0.5%) [8]. Hence, low order, second (CD2) and third (CD3) caesarean deliveries, comprise most of the repeat CDs (92%) [8]. The main challenge of lower order CD, assuming normal placental adherence, is abdominal-pelvic adhesions. Damage to the peritoneum, the protective surface layer of various abdominal-pelvic organs, during surgery initiates a physiological cascade that could lead to adhesion formation. This cascade is based on cytokines and other inflammatory mediators, released by the damaged mesothelial cells and macrophages, that stimulate mesothelial migration to the damaged area, thus, initiating the process of re-epithelialisation. As part of the inflammatory cascade, fibrin is deposited and provides a matrix that facilitates repair. During this process, two damaged adjacent peritoneal surfaces may join to form an adhesion. Normally during the healing process, tissue plasminogen activator (tPA) and urokinase plasminogen activator (uPA), found in mesothelial cells, enable the conversion of plasminogen to plasmin, which breaks the fibrin to fibrin-degradation products. Any reaction that interferes with this process, such as the excess of plasminogen activating inhibitor (PAI)-1 and PAI-2, can lead to adhesion formation [9]. Infection, hypoxia, tissue manipulation and chemical irritation are all potential factors for the activation of the abdominal-pelvic adhesions cascade leading to fibrous band-like structures that could potentially cause chronic pelvic pain, bowel obstruction, infertility, and complications in future surgeries [9,10]. During the process of adhesiolysis, in a subsequent surgery, damage to the adjacent muscles and structures such as, the ureter, bladder, and colon may occur, with a potential for massive bleeding/blood transfusion, peritonitis, infection, and septic shock. These complications can result in acute and chronic renal failure, need for a relaparotomy, and admission to the intensive care unit, resulting in prolonged maternal hospitalization [5–10]. Adhesiolysis increases the length of the operation and was hypothesized to contribute to the high rate of perioperative maternal complications [11–13]. Prolonged operative time was shown to be a risk marker for perioperative maternal complications and is associated with the presence of abdominal-pelvic adhesions [11,12]. Previous studies defined a complicated maternal caesarean delivery by a composite outcome including, excessive bleeding requiring blood transfusion, relaparotomy, uterine rupture/dehiscence, hysterectomy, injury to bladder/ureter or bowel, wound infection, endometritis/sepsis or prolonged operative time [10–13].

The purpose of this study was to examine the risk factors for a complicated maternal caesarean delivery (CD) among women undergoing a repeat low-order CD, and to develop a predictive model for women at risk. Specifically, we aimed to study if women who experienced a composite complicated maternal CD, unrelated to abnormal placentation, during the second caesarean delivery (CD2), were at increased risk for perioperative maternal complications in the subsequent CD (CD3), using a matched group of women. As data was collected from the same woman at two time points, each served as her own control.

## Materials and methods

A retrospective longitudinal study of all women who underwent two low-order repeat CDs (CD2 and CD3) at a single large university-affiliated, tertiary medical centre, Shaare Zedek

Medical Center (SZMC), Jerusalem, between July 2005 and December 2016. Obstetric surgical reports were extracted from the electronic medical record containing both mandatory fields and free text summary notes, continuously updated during labor and delivery. Maternal variables extracted from the mandatory fields included age, BMI, previous medical history, parity, number of previous vaginal and CDs, noting if performed in another facility, gestational age, indication for CD, duration of CD, length of interpregnancy interval, trial of labor, uterine rupture, uterine dehiscence, blood transfusion, gestational diabetes or hypertension. Fetal variables extracted included number of fetuses, gender, birth weight, Apgar score, and admission to the neonatal intensive care unit.

Surgical technique for all CDs performed at our facility was comprised of skin opening by Pfannenstiel's incision and blunt extension of uterine cut. However, among senior surgeons there existed slight variations with respect to fascia opening (blunt/sharp/electrosurgery), uterine closure (one versus two layers), closure versus non-closure of peritoneum and muscles and skin closure (staples/monocryl/nylon). The vast majority of repeat CDs (greater than 95%) were performed under regional anesthesia. Unique to our medical center, women with two documented prior low segment transverse CDs (LSTCD) were eligible for a trial of labor (TOLAC) under a strict protocol that includes the absence of previous uterine dehiscence or rupture, estimated fetal weight < 4000, vertex presentation, previous vaginal delivery and spontaneous progress of labor [14]. As such, in both CD2 and CD3 women may have had a trial of labor. The operative time of the CD was measured from the of the initial incision to skin closure and was a mandatory field in the electronic medical record, documented in all CD [10,11]. Prolonged operative time was defined as a CD that exceeded the 90th percentile of the operative time of the study group [10]. The outcome, composite maternal complicated CD, was comprised of at least one of the following: (1) a clinically significant hemorrhage necessitating blood transfusion, (2) a uterine rupture or dehiscence, (3) a relaparotomy for bleeding control, (4) an admission to the intensive care unit or (5) a prolonged operative time ≥ 90th percentile. Relaparotomy and uterine rupture/dehiscence were identified using International Classification of Diseases, 9th Revision, Clinical Modification Diagnosis Codes. Women with placenta accreta spectrum and those who underwent a hysterectomy were excluded from the study, since the intent was to focus on complications not related to abnormal placentation.

The primary outcome of the study was to evaluate if a complicated CD2 is associated with complications in the subsequent CD (CD3) and if such, to calculate the sensitivity (Sn), specificity (Sp), positive and negative predictive value (PPV, NPV) of a complicated CD2 on maternal complications in CD3. Secondary outcomes were to (1) identify risk factors at CD3 associated with a composite complicated maternal CD3, and to (2) build a statistical model including factors from CD2 and CD3 to predict women at risk for a composite maternal complication during CD3.

## Statistical analysis

Data was validated by defining distributions and quantifying missing values. Obstetric characteristics comparing CD2 and CD3 were presented as proportion, median or mean with interquartile range respectively or standard deviation, depending on the variable characteristics; categorical, ordinal or continuous, respectively. Statistical significance was defined by a two-sided p value ≤ 0.05 using the Chi-square test or Fisher Exact test for categorical variables. Wilcoxon Signed Rank Test was employed for ordinal or continuous variables with non-Gaussian distribution and the Student Paired t-test was utilized for continuous variables with a normal distribution. A univariate analysis was performed followed by the development of a multivariate logistic regression model to evaluate if a composite complicated maternal CD2 is

an independent risk factor for complications in the subsequent CD (CD3). The association between a composite complicated maternal CD2 and CD3 was evaluated by the Chi-square test. The Sn, Sp, PPV and NPV of a complicated CD2, for detecting complications in the subsequent CD3 was calculated.

## Sample size

Based on previous studies [9] the difference between CD2 and CD3 in respect to the prevalence of severe adhesions was 28% and 41% respectively. The sample size needed to find such a difference with a power of 80% was 418 (209 in each group). In an effort to decrease selection bias, the entire cohort that met inclusion criteria was analyzed. As the annual delivery volume is approximately 14,500 with a CD rate of 12%, we assumed an 12 year time frame would contain enough women needed for the study.

The study was performed in accordance with the ethical standards of the Declaration of Helsinki and its later amendments and was approved by the Institutional Review Board of the Shaare Zedek Medical Center (SZMC IRB 0260-16-SZMC). The manuscript is presented according to the STROBE guidelines [15].

## Results

During the study period, there were 159011 deliveries, 18695 (12%) underwent CD, 3974 (21%) were a recurrent CD of which 3305 (83%) were low order CD2 and CD3. We identified 1331 women eligible for inclusion in the study (Fig 1).

Women presenting for CD3 compared to their CD2 were an average of three years older, had a longer interpregnancy interval (7 months) and were less likely to attempt TOLAC (8% vs 35%, p<0.001). The operative time for CD3 was longer than CD2 by a mean of 3 minutes, with 142 (11%) of operations exceeding the 90th percentile (62 minutes), compared to 99 (7%) in CD2, p = 0.002.

As shown in Table 1, of the 1331 women in the cohort, 159 women (11.9%) had a composite complicated maternal CD2 compared to 226 (16.9%) of women during CD3, an increase of 67 cases (42%) p<0.001. Rate of maternal complications including uterine scar dehiscence/rupture, need for relaparotomy and blood transfusion, were similar for both groups. None of the women were admitted to the intensive care unit for postpartum observation or treatment at either time.

As shown in Table 2, the univariate analysis identified six factors significantly associated with an increased risk for a composite complicated maternal CD3. Factors related to CD2 included prolonged operative time (≥90th percentile, OR = 2.4 p<0.001), dehiscence or rupture or relaparotomy (OR = 2.1 p = 0.013), while Sub-Saharan African origin (OR = 3.2 p<0.001), gestational age (OR = 0.8 p<0.001), birth weight (OR = 0.9 P<0.001) and TOLAC (OR = 2.1 p<0.001) were related to CD3. A composite maternal complication at CD2 was consistent with an increased risk for a complicated CD3 (OR = 2.3 P<0.001).

A binomial logistic regression was performed to determine the effect of a maternal complication in previous CD2 together with potential risk factors at CD3 on the likelihood of composite maternal complication in CD3. The model included; maternal age, parity, preterm delivery, TOLAC and birthweight of newborn, a Sub-Saharan African origin and a composite maternal complication in previous CD2. The model included 1328 cases, was statistically significant, $\chi2(7) = 50.760$, p <0.001, explained 6.3% of the variance of composite complicated maternal CD3 and correctly classified 82.9% of cases (Table 3).

Sensitivity of the model was 1.8%, specificity was 99.5%, PPV and NPV were 44.4% and 83.1% respectively. Of the seven predictor variables, only three were statistically significant;

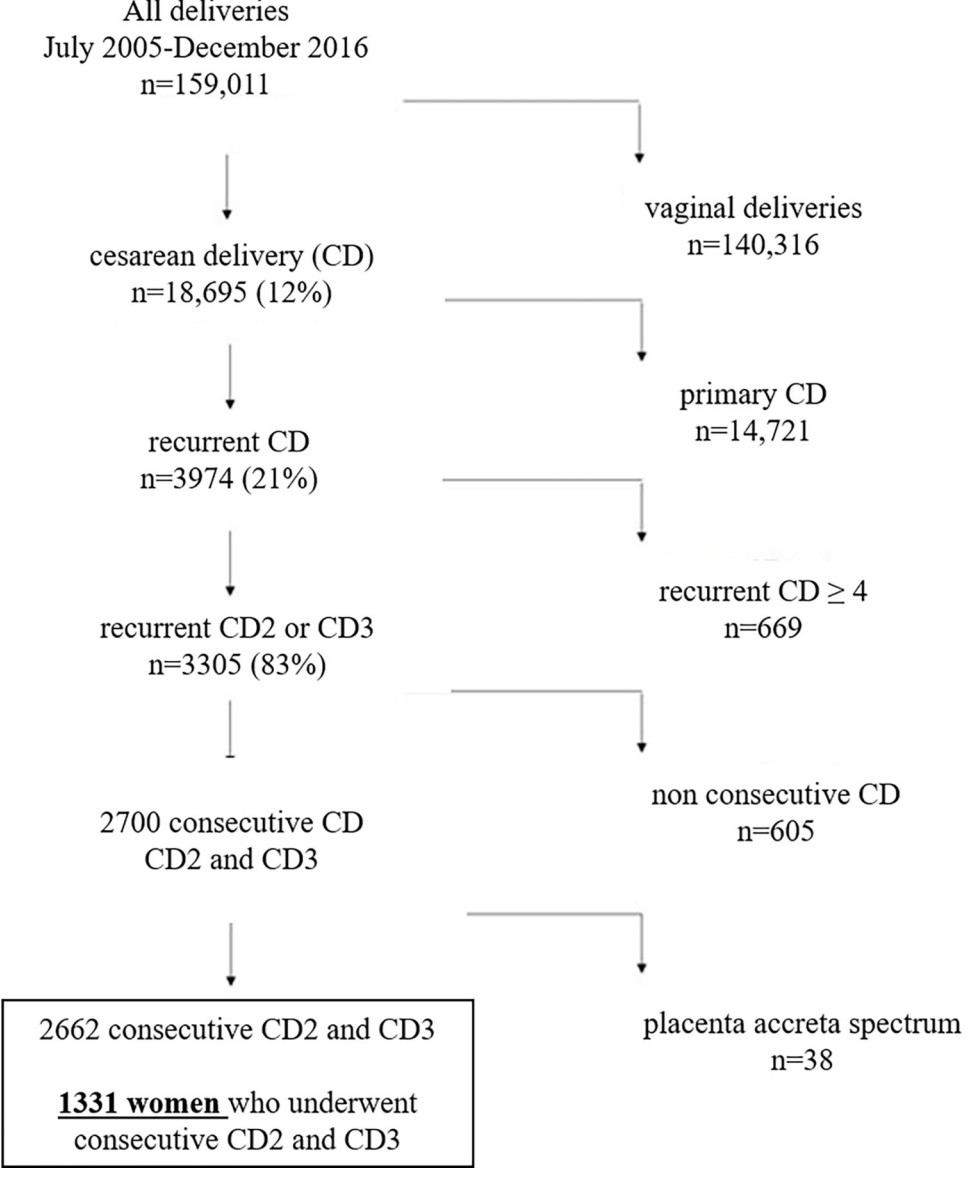

**Fig 1. Flow chart of study population.**

composite complicated maternal CD2 aOR = 2.2 p<0.001, preterm delivery aOR = 1.7 p = 0.010 and Sub-Saharan African origin aOR = 3.7 p<0.001. The area under the ROC curve was 0.639, 95% CI [0.597 0.681].

A composite complicated maternal CD2 was found to be associated with maternal complications in CD3, p < 0.001. The potential use of a "composite complicated maternal CD" as a "diagnostic tool" for maternal complications in CD3 was assessed. As noted in Table 4, the Sn, Sp, PPV, and NPV were 21%, 90%, 30% and 85% respectively.

## Discussion

There are limited studies that focus solely on repeat low order CD. The majority of the studies combine low and high order repeat CD in fixed groups, and some even include the primary

**Table 1. Obstetric characteristics of 1331 parturients undergoing repeat caesarean delivery (CD) comparing CD2 and CD3.**

| | Second CD (CD2) | Third CD (CD3) | P value |
|---|---|---|---|
| **Maternal characteristics / obstetrical history** | | | |
| Maternal age at delivery, years, mean (±sd) | 30 (±5) | 33 (±5) | <0.001 |
| Sub-Saharan African descent | | 39 (3%) | |
| Non Jewish | | 215 (16%) | |
| Maternal BMI, median (25th, 75th) | 29.5 (26, 36) | 30.4 (27, 30) | 0.579 |
| Parity, median (min., 25th, 75th, max) | 2 (2, 2, 4,16) | 3 (3, 3, 5, 17) | <0.001 |
| Gestational age at delivery, mean (± sd) | 38.5 (±2.1) | 37.4 (±1.6) | <0.001 |
| Interpregnancy interval, months, median (25th, 75th) | 18 (13, 27) | 25 (15, 36) | <0.001 |
| Gestational diabetes, N (%) | 126 (9%) | 146 (11%) | 0.224 |
| Hypertensive disorders, N (%) | 71 (5%) | 55 (4%) | 0.171 |
| Assisted Reproductive Technology, N (%) | 111 (8%) | 78 (6%) | 0.016 |
| Multiple pregnancy, N (%) | 53 (4%) | 25 (2%) | 0.003 |
| Preterm delivery <37 weeks, N (%) | 145 (11%) | 174 (13%) | 0.055 |
| Trial of labor after CD, N (%) | 463 (35%) | 106 (8%) | <0.001 |
| Vertex presentation | 1144 (86%) | 1193 (90%) | 0.01 |
| **Maternal complications** | | | |
| Uterine rupture, N (%) | 19 (1.4%) | 12 (0.9%) | 0.278 |
| Uterine scar dehiscence, N (%) | 36 (2.7%) | 50 (3.8%) | 0.154 |
| Hemoglobin drop ≥ 3g%, N (%) | 150 (11%) | 100 (8%) | 0.001 |
| Blood transfusion, N (%) | 41 (3%) | 51 (4%) | 0.341 |
| Duration of CD, minutes, mean (± sd) | 37 (±18) | 40 (±21) | <0.001 |
| Duration of CD > 90 percentile (> 62 min), N (%) | 99 (7%) | 142 (11%) | 0.002 |
| Relaparotomy, N (%) | 0 | 3 (0.2%) | 0.250 |
| Admission to the intensive care unit, N (%) | 0 | 0 | |
| Composite complicated maternal CD* | 159 (12%) | 226 (17%) | <0.001 |
| **Neonatal outcome** | | | |
| Birthweight, mean (± sd) | 3211 ±636 | 3091 ±544 | <0.001 |
| Macrosomia (>4000g) N, (%) | 115 (9%) | 52 (4%) | <0.001 |
| Apgar at 5 minutes ≤ 7 N, (%) | 42 (3%) | 35 (3%) | 0.488 |
| Neonatal intensive care unit N, (%) | 108 (8%) | 105 (8%) | 0.886 |

* Composite complicated maternal **CD** either one of the following: Blood transfusion > 1 packed cells, uterine rupture, dehiscence, relaparotomy or duration of operation > 90 percentile.

CD as a reference group, as such it is difficult to seek conclusions specifically on the group of low order repeat CD [8,13,16–20]. As summarized in Table 5, most studies were retrospective, compared between low and high order CD on various outcomes, including maternal complications, focusing mainly on the effect of the order of CD [8,13,16–19].

The current study appears to be the first longitudinal follow up study reported in the English literature of a cohort of women focusing exclusively on repeat low order CD, excluding women with placental accreta spectrum, aiming to predict maternal complications in CD3 based on specific parameters of CD2 and CD3. We found that a composite complicated maternal CD2 was associated with maternal complications in CD3, p < 0.001 (Table 4). Noteworthy, the fact that a women experienced a composite complicated maternal CD2 does not serve as a "diagnostic tool" for a complication in CD3 given that the Sn and PPV are low. Of the 226 cases with a composite complicated maternal CD3, only 47 (21%) experienced a complication

**Table 2. Univariate analysis evaluating potential risk factors for a composite maternal complication at CD3 including factors from the CD2 and factors from the studied CD3.**

| | Univariate OR (CI) | P value |
|---|---|---|
| **Factors of CD2** | | |
| CD2 Prolonged CD > 90 percentile | 2.4 (1.5; 3.7) | <0.001 |
| CD2 Dehiscence / rupture / relaparotomy | 2.1 (1.2; 3.9) | 0.013 |
| CD2 Blood transfusion | 1.8 (0.9; 3.7) | 0.093 |
| CD2 Composite complicated maternal CD2 | 2.3 (1.6; 3.4) | <0.001 |
| **Factors of CD3** | | |
| CD3 Sub-Saharan African origin | 3.2 (1.6; 6.2) | <0.001 |
| CD3 Non-Jewish | 1.3 (0.8; 1.8) | 0.138 |
| CD3 Age (maternal) | 1.0 (1.0; 1.0) * | 0.839 |
| CD3 BMI | 1.0 (0.9; 1.0) ** | 0.474 |
| CD3 Parity | 1.0 (0.9; 1.0)# | 0.831 |
| CD3 Gestational age at delivery | 0.8 (0.8; 0.9) | <0.001 |
| CD3 Gestational age at delivery & Composite complicated maternal CD2 | 1.0 (1.0–1.0) ## | <0.001 |
| CD3 Birthweight (newborn) | 1.0 (1.0–1.0)### | <0.001 |
| CD3 Trial of labor | 2.1 (1.4.;3.3,) | <0.001 |
| CD3 Trial of labor & Composite complicated maternal CD2 | 1.6 (0.4–6.1) | 0.462 |
| CD3 Hypertensive disorders | 1.2 (0.6;2.3) | 0.586 |
| CD3 Gestational diabetes | 0.8 ((0.5;1.3) | 0.410 |
| CD3 Nonvertex | 0.7 (0. 5;1.1,) | 0.743 |
| CD3 Interpregnancy interval | 1.0 (1.0;1.0) ^ | 0.179 |

Composite complicated maternal CD either prolonged operative time, blood transfusion, dehiscence/uterine rupture or relaparotomy.

*1.006(0.979;1.034).

**0.978(0.935;1.023).

#1.007(0.946;1.072).

##1.023 (1.012–1.033).

###0.999 (0.999;1.000).

^1.007(0.997;1.017).

**Table 3. Logistic regression predicting likelihood for a composite complicated maternal CD3.**

| | B | S.E. | Wald | df | Sig. | Exp(B) | 95% C.I.for EXP(B) | |
|---|---|---|---|---|---|---|---|---|
| **Factors from previous CD2** | | | | | | | | |
| Composite complicated maternal CD2 | 0.827 | 0.196 | 17.779 | 1 | <0.001 | 2.286 | 1.556 | 3.355 |
| **Factors from CD3** | | | | | | | | |
| Age | -0.01 | 0.017 | 0.006 | 1 | 0.939 | 0.999 | 0.966 | 1.032 |
| Parity | 0.22 | 0.038 | 0.346 | 1 | 0.556 | 1.023 | 0.949 | 1.102 |
| Preterm delivery <37 gestational week | 0.572 | 0.223 | 6.566 | 1 | 0.010 | 1.772 | 1.144 | 2.745 |
| TOLAC | 0.407 | 0.252 | 2.620 | 1 | 0.105 | 1.503 | 0.918 | 2.460 |
| Birthweight | 0.000 | 0.000 | 2.297 | 1 | 0.130 | 1.000 | 0.999 | 1.000 |
| Sub-Saharan African descent | 1.312 | 0.348 | 14.208 | 1 | <0.001 | 3.713 | 1.877 | 7.346 |

Included in the model 1328 women, $\chi2(7) = 50.760$, $p < .001$. Nagelkerke R Square = 0.063, Hosmer and Lemeshow = 0.427.

**Table 4. The association between a composite complicated maternal CD2 and a composite complicated maternal CD3.**

| | | composite complicated maternal CD3 | | |
| --- | --- | --- | --- | --- |
| | | yes | no | total |
| composite complicated maternal CD2 | yes | 47 | 112 | 159 |
| | no | 179 | 993 | 1172 |
| | total | 226 | 1105 | 1331 |

Person Chi-Square < 0.001.

Composite complicated maternal CD defined as either one of the following: Blood transfusion > 1 packed cell, uterine rupture, dehiscence, relaparotomy or duration of operation > 90 percentile.

in the previous CD Sn = 21% and of the 159 women with a composite complicated maternal CD2, 112 (70%) were not complicated in CD3, PPV = 30%. However of the 1172 who had an un-eventful CD2, only 179 experienced complication in CD3 (15%). The NPV of a complicated CD2 as a diagnostic tool for identifying women with a complicated CD3 was 85%.

Initially we developed a multivariate logistic regression model including maternal characteristics such as age, parity, gestational age, BMI, together with the significant factors identified by the univariate analysis. Given that BMI only became a required field in the medical record in the last three years of the study, it was absent in 68% of cases of CD3. Therefore, we developed three different binomial logistic regression models. The first model included BMI, age, gestational age, parity, TOLAC, birth weight, Sub-Saharan African origin and a composite complicated maternal CD2. The second model contained all the variables in the first model

**Table 5. Studies evaluating maternal complications among women undergoing repeat CD.**

| Reference number | Methodology | Aim of the study | Order of CD / Groups compared | Placenta accreta spectrum | Maternal complications |
| --- | --- | --- | --- | --- | --- |
| 8 | Prospective observational cohort | descriptive: comparing high order to low order CD | CD1/ CD2 / CD3 / CD4 / CD5 / ≥ CD6 elective CD | included | dehiscence / rupture / blood transfusion / bowl-bladder injury / cesarean hysterectomy/ intensive care unit admission |
| 13 | Retrospective descriptive study. | descriptive: comparing high order to low order CD | CD1/ CD2 / CD3 / ≥ CD4 elective CD | included | dehiscence / rupture / blood transfusion / bowl-bladder injury / wound infection/ UTI/ DVT/ cesarean hysterectomy/ duration of surgery |
| 16 | Retrospective descriptive study. | descriptive: comparing high order to low order CD | CD1/ CD2 / ≥ CD3 elective and emergent CD | included | dehiscence / rupture / blood transfusion / bowl-bladder injury / wound infection/ UTI/ DVT/ cesarean hysterectomy/ wound infection |
| 17 | Retrospective descriptive study. | descriptive: comparing high order to low order CD logistic regression model for bladder injury (2 cases) bowel injury (1 case) cesarean hysterectomy (2 cases) | CD2 / ≥ CD3 elective and emergent CD | included | dehiscence / rupture / blood transfusion / bowl-bladder injury / wound infection/ UTI/ DVT/ cesarean hysterectomy/ duration of surgery |
| 18 | Retrospective descriptive study. | descriptive: comparing high order to low order CD | CD2 & CD3 / ≥ CD4 elective CD | included | dehiscence / rupture / blood transfusion / bowl-bladder injury / cesarean hysterectomy/ duration of surgery |
| 19 | Retrospective descriptive study. | descriptive: comparing high order to low order CD in regard to the timing of repeated CD after two or more previous CD | CD2 / ≥ CD3 | included | dehiscence / rupture / blood transfusion / bowl-bladder injury / cesarean hysterectomy/ wound complication/ DVT |

except for BMI. The third model, similar to the first model, but instead of a composite complicated maternal CD2, each one of the complications at CD2 was added independently to the model. Comparing the three models with regards to the number of cases included in the model, significance and Nagelkerke R Square (NRS) showed superiority to Model 2. (Model 1; n = 421, p = 0.061 and NRS = 0.058, Model 2; n = 1328, p<0.001 and NRS = 0.063, Model 3; n = 420, p = 0.194 and NRS = 0.058). Of the risk factors identified in the univariate analysis, only three remained significant in the multivariate logistic regression model; composite complicated maternal CD2 aOR = 2.2 p<0.001, preterm delivery aOR = 1.7 p = 0.010 and Sub-Saharan African origin aOR = a3.7 p<0.001. Although the model was significant and identified independent risk factors for a composite complicated maternal CD3, the area under the ROC curve was 0.639, 95% CI (0.597 0.681), indicating a poor prognostic capability of the model.

Among the group of low order CDs, adhesiolysis is a major risk factor for maternal perioperative complications [10]. Various risk factors for developing postsurgical pelvic/abdominal adhesions are recognized including infection, tissue ischemia, excess manipulation of tissues, intraperitoneal bleed, dissection of previous adhesions and presence of reactive foreign bodies [9,10,21–23]. Most importantly, the prevalence of adhesions increases with the order of CD, as severe adhesions were present in 28% of CD2, 41% of CD3 and 50% of CD4 [13]. It is assumed that genetic variations between patients can contribute to adhesion formation, yet a specific genetic culprit has not been identified [23]. A small percentage of the study population (n = 39, 3%,) were of Sub-Saharan African descent (Ethiopian), who had an aOR of 3.7 for a composite complicated maternal CD3 compared to Caucasian women. This is, to some extent, similar to the association found between the presence of keloids/dense pelvic adhesions in African American women [23].

There is increasing data regarding the role of microRNAs in cell development, differentiation, and proliferation. This family of small noncoding RNAs, range from 22 nucleotides in length, regulate gene expression. Their role in many human diseases including adhesion formation, preeclampsia and intrauterine growth restriction (IUGR) has been demonstrated [24,25]. Genetic variations between patients and ethnic origin could be attributed in part to variations in microRNAs. The search for an association between biomarkers or sonographic findings with specific diseases or pathological conditions including preeclampsia, stillbirth and IUGR is underway [26,27]. Studies have shown an association between specific biomarkers and severity of adhesions in endometriosis and sonographic findings associated with the severity of adhesions in repeat CDs [28,29]. Future studies focusing on a complication of repeat CD could combine these factors with the clinical risk factors identified in the current study.

There is a strong association between length of operation and presence of adhesions, adhesiolysis and maternal complications [11–13]. In the current study previous prolonged operative time (> 90th percentile at CD2) increased risk for a maternal complication in CD3 by an OR of 2.4, p<0.001. When applying each one of the complications of CD2 independently, instead of as a composite, (logistical regression Model 3), the predictor variable of prolonged operative time at CD2 remained statistically significant (aOR of 2.6, p = 0.013). As such, operative time is an important risk factor for a maternal complication in the subsequent CD. Since operative time is a mandatory field in most electronic medical records and its accessibility make it a practical and important factor available for use when counseling women about the risk of complications in future CDs.

This study retrieved data across the span of 12 years (2005–2016). During this time period there have been changes to surgical practice, the development of materials and medications to aid in hemostats, an increase in the technology and uptake of fertility treatments and social factors, including postponing pregnancy, all with a potential to effect the study outcome. To evaluate if the years of the study were associated with a composite complicated maternal CD, we

divided the cohort into two groups, women who delivered both CD2 and CD3 in the "early years" 2005–2011 (n = 384) and those who delivered both CD2 and CD3 during the "late years", 2012–2016 (n = 462). Women who had the first CD2 at "early years" and CD3 at "late years" were excluded (n = 485). We found a significant association between a composite complicated maternal CD2 versus CD3 in "early years" 2005–2011, 34 (8.9%) versus 47 (12.2%), and in late years 66 (14.3%) versus 85 (18.4%), p = 0.015. To evaluate this effect on the model, "early" and "late" years were entered into the logistic regression model as a categorical variable. However, this addition did not significantly alter the previous reported aOR for the various predictors in the model. Although significance was not reached for the variable of "early" versus "late" years (aOR = 1.5, p = 0.054), being close to significance suggests a need to address this issue in future studies.

Strengths of the current study; We controlled for confounders by (1) including only low order CD, (2) performing a multivariate logistic regression analysis (3) having each woman serve as her own control (as CD2 and CD3 were matched cases) and thus, many of the known and unknown confounding factors that were not available or missing from the dataset, were controlled for. Examples for such factors include socioeconomic status, smoking status, alcohol intake, and genetic factors influencing tissue repair. (4) Of the 24 variables retrieved from the electronic medical record, 16 (66%) were from mandatory fields, thereby minimizing missing data. (5) Given that the study was performed in one clinical center, there was limited significant variability in the surgical technique. (6) The similarity between the results of the current study and the large multicenter prospective, observational cohort from the United States regarding rates of blood transfusion and time of operation, strengthens the external validity of this study.[6]

Limitations of the study; (1) Single center studies by nature are characterized by a more homogenous population as compared to multicenter studies and have potential differences in obstetric management and treatment protocols which could affect the external validity. Given the multifactorial potential factors for complications of repeat CD (surgical techniques, various hemostatic materials, ethnic groups, maternal health conditions as diabetes and obesity), the generalization of the study findings should be with caution. (2) BMI became a mandatory field in the medical record in the last three years, therefore, it was missing in 68% of cases. However, given that every woman was her own control, and assuming that BMI was similar between CD2 and CD3, the effect on CD2 and CD3 would have been similar. (3) Both diagnosis of 'adhesiolysis' and 'injury to internal organs' are not mandatory fields in the electronic medical record. We assumed that surgeries complicated by excessive adhesiolysis and those with injury to internal organs would be included in the cases of 'prolonged operative time'.

## Conclusions

Based on the results of this study, we suggest that a composite complicated maternal CD2 together with a Sub-Saharan African origin and a preterm delivery in the subsequent CD3 are all risk factors for maternal complications in CD3. Although it is hard to predict which women will experience complications, it is easier to predict which woman will undergo an uneventful CD3 given the SP of 99.5% and the NPV of 83% of the model. Identifying women who are not at risk for complication may enable us to provide better counselling for women undergoing a repeated low order CD.

## Supporting information

**S1 File.**
(SAV)

## Author Contributions

**Investigation:** Orna Reichman.

**Methodology:** Orna Reichman.

**Supervision:** Orna Reichman, Hen Y. Sela, Sorina Grisaru-Granovsky.

**Writing – original draft:** Orna Reichman, Misgav Rottenstreich.

**Writing – review & editing:** Orna Reichman, Misgav Rottenstreich, Hen Y. Sela, Rachel Michaelson-Cohen, Zvi Ehrlich, Reut Rotem, Sorina Grisaru-Granovsky.

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
