## [Decision Letter · Decision Letter 0]

14 Aug 2022

PONE-D-22-18931Repeat low order caesarean delivery, risk factors for complications; a retrospective longitudinal study.PLOS ONE

Dear Dr. Reichman,

Thank you for submitting your manuscript to PLOS ONE. After careful consideration, we feel that it has merit but does not fully meet PLOS ONE’s publication criteria as it currently stands. Therefore, we invite you to submit a revised version of the manuscript that addresses the points raised during the review process.

We look forward to receiving your revised manuscript.

Kind regards,

Antonio Simone Laganà, M.D., Ph.D.

Academic Editor

PLOS ONE

Journal Requirements:

2. In your Methods section, please state at which hospital the study was conducted.

Additional Editor Comments:

The topic of the manuscript is interesting. Nevertheless, the reviewers raised several concerns: considering this point, I invite authors to perform the required major revisions.

Reviewers' comments:

Reviewer's Responses to Questions

**Comments to the Author**

1. Is the manuscript technically sound, and do the data support the conclusions?

Reviewer #1: Partly

Reviewer #2: Yes

2. Has the statistical analysis been performed appropriately and rigorously? 

Reviewer #1: No

Reviewer #2: Yes

3. Have the authors made all data underlying the findings in their manuscript fully available?

Reviewer #1: No

Reviewer #2: Yes

4. Is the manuscript presented in an intelligible fashion and written in standard English?

Reviewer #1: Yes

Reviewer #2: Yes

5. Review Comments to the Author

Reviewer #1: General

• In this paper by Reichman, et al. the authors have conducted a retrospective study aimed at assessing a complicated 2nd caesarean surgery on the proababilty of a complicated 3rd caesarean surgery.

• In regard to language, the paper in its current form is understandable. However, I do believe that the manuscript will significantly benefit from language counselling, in terms of readability as well as clarity.

• While the included cohort is comprehensive and impressive, allowing for an impressive statistical power, there are still some questions that need to be asked regarding the statistical methods applied.

• While signing data availability, I could not locate any mention of supposedly available data.

• Lastly, the manuscript includes a total of 19 references with 10 of them (52%) being from the last decade. It is clear that the authors have chosen to refer to updated and high-quality literature in this manuscript as seen in the high percentage of journal from the last 5 years and from the last decade.

Introduction

• The introduction section of the manuscript gives sufficient information to understand the challenge and the clinical-scientific gaps between literature and the focus of the manuscript.

• However, while the authors eloquently described the implications of adhesiolysis such as possible damages to adjacent tissues and prolonged surgery, the fail to explain the process and approaches by which adhesiolysis is achieved. Given that Plos One is a wide-spectrum journal, I would suggest some elaboration on that part in order to give context to your findings and challenges.

• The last paragraph of the introduction is cumbersome and could be better phrased if the authors are interested in helping the readers understand their aim.

Materials and Method

• The methods section of the manuscript is clear, concise, and yet comprehensive. Although power calculations were performed for a specific complication, the calculation emphasizes that the study is well-powered for the declared aim.

• The authors state that the study included women who underwent two consecutive low order repeats CDs in their tertiary medical center. Were there any measures taken to assess the possibility of TOLAC or CD in another medical center in the period between the documents CDs in your center? If so, please state it in the manuscript.

• Do women who experience significant complications such as rupture or dehiscence in CD2 are allowed to attempt TOLAC in your institute? If so, were they flagged in your DB?

• Regarding maternal complications, the authors state excessive hemorrhage treated with two or more packed red blood cells transfusions. Why did the authors exclude cases treated with 1 packed red blood cells does or alternatively, patients who have bled but prior hemoglobin concentrations were high enough to avoid the administration of packed red blood cells?

Please consider expanding your maternal complication definition to include postsurgical hemoglobin drop (as you’ve already measured it and state it table 1).

• As the cohort this study evaluates was operated on over a period of decade, surgical techniques and tools as well monitoring techniques and general opinion and approach to medical care might have also changed. Furthermore, fertility treatments have significantly improved over this period, which may have lead to increased maternal ages in later periods of the cohort. Have the authors evaluated low order CD complication rates in women from the both beginning and end of the study period?

• Regarding statistical analysis methods:

o Some assessed variables (e.g. parity, BMI) are characterized by non-gaussian distribution and would therefore require the application of non-parametric tests.

o It is not clear from the text if the authors have used matched-samples t-test or not. Given the matched nature of this study, please make sure that the proper test was used and amend the text accordingly.

o How did you assess the parametric or non-parametric nature of some of the variables? (e.g. length of interpregnancy interval).

o Please state what statistical test was performed for each variable.

o Were intervariable interactions assessed before introduction to the multivariate model?

Results

• The flow chart provided by the authors greatly assists in understating the cohort composition. However, it is not clear what are the reasons for the exclusion of 1350 (50.4%) women of the 2781 women who have underwent consecutive CD2 and CD3 without placenta accreta. Please elaborate on the exclusion reasons and provide a matching table.

• Table 1:

o Please state if mean of median was used in BMI.

o Given that your institution is also caring for grandmultiparity cases, please use interquartile range when using a median (e.g. parity).

o Please state the gestational week of the pregnancies.

o Please include all assessed variables relevant for the cohort characterization (even if non-comparable) such as ethnicity, religion and comorbidities. Especially so if those variables are later used in the multivariate model.

• Page 5, line 132 – I believe that () marks were supposed to be placed around the TOLAC data.

• Table 2:

o Please include the p-value for each assessed variable.

o Please make sure that the multivariate model includes BMI, maternal age, gestational age and parity as adjusting variables.

o Please consider reporting the final amount of cases included in the model, its p-value and -2LL.

o This is a suggestion only: consider establishing a second model in which the specific association of any complication in CD2 with a complication in CD3, adjusted only to BMI, maternal age, gestational age and parity. I believe that while the current model is informative, the suggested model will be easier to apply in clinic and will answer the aims of the study better.

In this context, please note that the current model has

• Page 6, lines 143-148 – Please refrain from stating table 2 in the end of each sentence.

• While it is easy to deduce the percentage of non-complicated and complicated cases in CD2 and CD3, it is not as easy to deduce the overlap between the groups. Please provide a 2x2 table to allow for easier calculation of sensitivity, specificity, positive and negative predictive values.

Discussion

General notes:

• The discussion chapter of this manuscript in its current form could benefit significantly from language editing as well as self-proofing. While the general concepts the authors wish to convey are presented in this chapter, they are difficult to deduce and extract.

• While the reference list of this manuscript is updated and encompasses quotations from well-established journals, reference to those results in this chapter are lacking. By the end of the chapter one may still fail in understanding how do these results fit what is currently known about low-order CDs.

Furthermore, it is somewhat surprising no to find a reference to similar papers who have assessed low-order CDs in the past (e.g. PMID 16501678, 30463461)

• I would assume that the multivariate model in its current form will alter after the introduction of the suggested variables. Please revise the data in this segment of the manuscript as well.

• In the perspective of this reviewer, the strongest aspect of this manuscript lays in its cohort size and multivariate logistic regression model. However, this model is poorly referred to in the discussion, with specific odds ratios not being mentioned.

In addition to that, this section of the manuscript requires some more specific alterations:

• Page 6 line 153 – I would suggest using a absolute numbers when stating a 40% increase as opposed to 17% of 12% (which is calculated to be 41.6%).

• Page 6 line 155 – it is unclear what do the authors mean by Viz.

• Page 6 lines 155-156 – The use of PPV and NPV is somewhat inaccurate in the context of prediction. It can be used in reference to an already established database to evaluate the accuracy of using a test (in this case, complications at CD2) to predict an outcome (in this case, complications at CD3). It is less accepted as a predictive tool. Please amend the text accordingly.

• Page 7 lines 173-175 – if I understand correctly, this group had a threefold risk for maternal complicated CD3. Please correct the text if this is correct.

• Strengths – It is suggested that the authors include limited variability in surgical technique, as expected of a one center study, as another strength of this study.

Conclusion

• The discussion barely refers to the difference in surgical duration and even tat only at the limitations segment. Therefore, it is not clear why the authors have decided to include the time of previous caesarean delivery as one of the important factors when planning a repeat CD. Either elaborate further on this finding, how it refers to currently existing knowledge and literature and why should it be interpreted the way you think it does, or omit it from the conclusion please.

Reviewer #2: I read with great interest the Manuscript titled “Repeat low order caesarean delivery, risk factors for complications; a retrospective

longitudinal study” (PONE-D-22-18931), which falls within the aim of this Journal.

In my honest opinion, the topic is interesting enough to attract the readers’ attention. Methodology is accurate and conclusions are supported by the data analysis. Nevertheless, authors should clarify some point and improve the discussion citing relevant and novel key articles about the topic.

Authors should consider the following recommendations:

- Manuscript should be further revised by a native English speaker.

- Does this manuscript conform the Enhancing the QUAlity and Transparency Of health Research (EQUATOR) network guidelines? It would be mandatory to declare about this element.

- Recent and novel evidence suggested that epigenetic changes, in particular altered expression of selective miRNA, may play a key role in both placental-induced diseases such intrauterine growth restriction. It would be mandatory to discuss (at least briefly) this topic, referring to: PMID: 28466013; PMID: 20104830.

- The real challenge in the era of molecular medicine is to find a biomarker, or even better a panel of biomarkers, for early diagnosis of pre-eclampsia, intrauterine growth restriction and stillbirth. I would stress this point, referring to: PMID: 28243732; PMID: 35245721.

6. PLOS authors have the option to publish the peer review history of their article (what does this mean?). If published, this will include your full peer review and any attached files.

Reviewer #1: No

Reviewer #2: **Yes: **Pietro Serra

---

## [Author Response · Author response to Decision Letter 0]

23 Sep 2022

September 22, 2022

Antonio Simone Laganà, M.D., Ph.D.

Academic Editor

PLOS ONE

Dear Prof, Lagana,

Thank you for the opportunity to revise this manuscript and resubmit it. The suggestions that the editors and reviewers have offered have been incorporated into the manuscript and have contributed greatly to the improvement of its quality. Based on the reviewers suggestions, major changes have been made to the manuscript including reanalyzing the data and rewriting the results and discussion sections. We don’t take it for granted the opportunity to resubmit a revised version and hope it is now appropriate for publication Below you will find our responses to each comment raised. 

Response: The manuscript was revised according to the PLOS ONE’s style guide. 

2. In your Methods section, please state at which hospital the study was conducted.

Answer: We added to the methods the hospital where the study was conducted:

"…A retrospective longitudinal study of all women who underwent two consecutive low order repeat CD, CD2 and CD3, in a single large university-affiliated, tertiary medical centre, at Shaare Zedek Medical Center, Jerusalem, between July 2005 and December 2016…"

Response: At the time of submission, we were not sure that we had permission to upload the data. After confirming with the Ethics Committee at our facility that it is possible to upload the data providing that the details are unidentifiable. We have given our approval to the journal and uploaded the data as a Supporting Information file.

Reviewer #1: General

• In regard to language, the paper in its current form is understandable. However, I do believe that the manuscript will significantly benefit from language counselling, in terms of readability as well as clarity.

Response: A professional, native English speaker language editor has reviewed and revised the manuscript.

• While the included cohort is comprehensive and impressive, allowing for an impressive statistical power, there are still some questions that need to be asked regarding the statistical methods applied.

• While signing data availability, I could not locate any mention of supposedly available data.

Response: At the time of submission, we were not sure if we had permission to upload the data. After confirming with the Ethics Committee at our facility that it is possible to upload the data providing that the details are unidentifiable. We have given our approval to the journal and uploaded the data.

• Lastly, the manuscript includes a total of 19 references with 10 of them (52%) being from the last decade. It is clear that the authors have chosen to refer to updated and high-quality literature in this manuscript as seen in the high percentage of journal from the last 5 years and from the last decade.

Response: Thank you

Introduction

• The introduction section of the manuscript gives sufficient information to understand the challenge and the clinical-scientific gaps between literature and the focus of the manuscript.

• However, while the authors eloquently described the implications of adhesiolysis such as possible damages to adjacent tissues and prolonged surgery, the fail to explain the process and approaches by which adhesiolysis is achieved. Given that Plos One is a wide-spectrum journal, I would suggest some elaboration on that part in order to give context to your findings and challenges.

Response: The flowing section was added to the introduction:

Line 66 – 81 "…The main challenge in this group of the lower order CDs, assuming normal placental adherence, is the presence of abdominal-pelvic adhesions. Damage to the peritoneum; the protective surface layer of various abdominal-pelvic organs during surgery initiates a physiological cascade which could lead to adhesion formation. This cascade is based on cytokines and other inflammatory mediators, released by the damaged mesothelial cells and macrophages that stimulate mesothelial migration, to the damaged area, initiating the process of re-epithelialisation. As part of the inflammatory cascade, fibrin is deposited and provides a matrix that facilitates the repair. During this process, two damaged adjacent peritoneal surfaces may join to form an adhesion. Normally during the healing process, tissue plasminogen activator (tPA) and urokinase plasminogen activator (uPA), which are found in mesothelial cells enable the conversion of plasminogen to plasmin, which breaks the fibrin to fibrin-degradation products. Any reaction that interferes with this process as the presence of excess of as plasminogen activating inhibitor (PAI)-1 and PAI-2 can lead to adhesion formation. [Surgical adhesions: A timely update, a great challenge for the future Andrew K. Davey, PhD and Peter J. Maher, MD ] Infection, hypoxia, tissue manipulation and chemical irritation are all potential factors for initiating abdominal-pelvic adhesions cascade leading to fibrous band like structures that potentially can lead to chronic pelvic pain, bowel obstruction, infertility and complications in future surgeries."

• The last paragraph of the introduction is cumbersome and could be better phrased if the authors are interested in helping the readers understand their aim.

Response: The last paragraph of the introduction was revised to clarify the aims of the study.

Line 95-102 “The purpose of this study was to study the risk factors for a complicated maternal caesarean delivery (CD) among women undergoing a repeat low-order CD, and to develop a predictive model for women at risk. Specifically, we aimed to study if women who experienced a composite complicated maternal CD, unrelated to abnormal placentation, during the second caesarean delivery (CD2), were at increased risk for perioperative maternal complications in the subsequent CD (CD3), using a matched group of women. Since data was collected from the same woman at two-time points, each served as her own control.”

Materials and Method

• The methods section of the manuscript is clear, concise, and yet comprehensive. Although power calculations were performed for a specific complication, the calculation emphasizes that the study is well-powered for the declared aim.

Thank you

• The authors state that the study included women who underwent two consecutive low order repeats CDs in their tertiary medical center. Were there any measures taken to assess the possibility of TOLAC or CD in another medical center in the period between the documents CDs in your center? If so, please state it in the manuscript.

Response: The electronic medical record includes mandatory data fields including number of previous vaginal and cesarean deliveries including if performed in another facility… We added to the method section the following paragraph.

Line 107-115 “Obstetric surgical reports were extracted from the electronic medical record containing both mandatory fields and free text summary notes, continuously updated during labor and delivery. Maternal variables extracted from the mandatory fields included age, BMI, previous medical history, parity, number of previous vaginal and CDs, noting if performed in another facility, gestational age, indication for CD, duration of CD, length of interpregnancy interval, trial of labor, uterine rupture, uterine dehiscence, blood transfusion, gestational diabetes or hypertension. Fetal variables extracted included number of fetuses, gender, birth weight, Apgar score, and admission to the neonatal intensive care unit.” 

• Do women who experience significant complications such as rupture or dehiscence in CD2 are allowed to attempt TOLAC in your institute? If so, were they flagged in your DB?

Response: Thank you for the comment and we have added the following text: 

Line 122-126 "Unique to our medical center, women with two documented prior low segment transverse CDs (LSTCD) are eligible for a trial of labor (TOLAC) under a strict protocol that includes the absence of previous uterine dehiscence or rupture, estimated fetal weight < 4000, vertex presentation, previous vaginal delivery and spontaneous progress of labor [14]. As such, in both CD2 and CD3 women may have had a trial of labor. ."

We were able to recognize from the data from women with uterine rupture or dehiscence as presented in Table 1

P value Third CD (CD3) Second CD (CD2) 

 Maternal complications

0.278 12 (0.9%) 19 (1.4%) Uterine rupture N (%)

0.154 50 (3.8%) 36 (2.7%) Uterine scar dehiscence N (%)

• Regarding maternal complications, the authors state excessive hemorrhage treated with two or more packed red blood cells transfusions. Why did the authors exclude cases treated with 1 packed red blood cells does or alternatively, patients who have bled but prior hemoglobin concentrations were high enough to avoid the administration of packed red blood cells?

Please consider expanding your maternal complication definition to include postsurgical hemoglobin drop (as you’ve already measured it and state it table 1).

Response: We thank the reviewer for his comment. This study’s aim was to focus on clinically significant maternal complications, as such we initially decided to base the definition of postpartum hemorrhage on women treated with at least two packed cells blood transfusion and not based on laboratory drop of HGB.

We agree with the reviewer’s comment regarding the need to expand and include women treated with one unit of packed cells (PC). We re-analyzed the data adding the 14 women who received only one unit of PC to the initial 78 women who received 2 or more PC. We included these women in the group of 'composite complicated maternal CD' and repeated all statistical analyses as presented in Table 1,2,3. 

In addition, as the reviewer suggested, we recalculated the results defining a composite maternal complication using a laboratory definition of a drop of HGB of more than 3 g% instead of need for a blood transfusion, as the reviewer can see below the findings were non-significant. We hope that the reviewer finds it acceptable that we defined PPH using the clinical definition and not the laboratory one. 

P value Third CD (CD3) Second CD (CD2) 

 Maternal complications

0.377 267 (20%) 248 (19%) *Composite outcome 

* Complicated CD either one of the following: HGB drop>=3g%, uterine rupture, dehiscence, relaparotomy or duration of operation > 90 percentile

• As the cohort this study evaluates was operated on over a period of decade, surgical techniques and tools as well monitoring techniques and general opinion and approach to medical care might have also changed. Furthermore, fertility treatments have significantly improved over this period, which may have lead to increased maternal ages in later periods of the cohort. Have the authors evaluated low order CD complication rates in women from the both beginning and end of the study period?

Response: We created a new variable dividing the cohort into:

women who delivered both CD2 and CD3 at "early years" 2005-2011 

women who delivered both CD2 and CD3 at "late years" 2012-2017

 and excluded the women who had the first CD2 at "early years" and CD3 at "late years" there was a significant difference between early and late years

The following section was added to the discussion section: 

Line 298-314 “This study retrieved data across the span of 12 years (2005-2016). During this time period there have been changes to surgical practice, the development of materials and medications to aid in hemostats, an increase in the technology and uptake of fertility treatments and social factors, including postponing pregnancy, all with a potential to effect the study outcome. To evaluate if the years of the study were associated with a composite complicated maternal CD, we divided the cohort into two groups, women who delivered both CD2 and CD3 in the "early years" 2005-2011" (n=384) and those who delivered both CD2 and CD3 during the "late years", 2012-2016 (n=462). Women who had the first CD2 at "early years" and CD3 at "late years" were excluded (n=485). We found a significant association between a composite complicated maternal CD2 versus CD3 in "early years" 2005-2011, 34 (8.9%) versus 47 (12.2%), and in late years 66 (14.3%) versus 85 (18.4%), p=0.015. To evaluate this effect on the model, "early" and "late" years were entered into the logistic regression model as a categorical variable. However, this addition did not significantly alter the previous reported aOR for the various predictors in the model. Although significance was not reached for the variable of "early" versus "late" years (aOR= 1.5, p=0.054), being close to significance suggests a need to address this issue in future studies.”

• Regarding statistical analysis methods:

o Some assessed variables (e.g. parity, BMI) are characterized by non-gaussian distribution and would therefore require the application of non-parametric tests.

Response : We have re-analysed the distribution of all variables using the P-P plot and the skewness and kurtosis. Ordinal and continuous variables with non-gaussian distribution were analysed by the Wilcoxon Signed Rank Test. These included: Parity, BMI, interpregnancy interval (months) and length of cesarean delivery.

o It is not clear from the text if the authors have used matched-samples t-test or not. Given the matched nature of this study, please make sure that the proper test was used and amend the text accordingly.

Response: We added to the method section:

Line 147-154 “Data was validated by defining distributions and quantifying missing values. Obstetric characteristics comparing CD2 and CD3 were presented as proportion, median or mean with interquartile range respectively or standard deviation, depending on the variable characteristics; categorical, ordinal or continuous, respectively. Statistical significance was defined by a two-sided p value ≤ 0.05 using the Chi-square test or Fisher Exact test for categorical variables. Wilcoxon Signed Rank Test was employed for ordinal or continuous variables with non-Gaussian distribution and the Student Paired t-test was utilized for continuous variables with a normal distribution.”

o How did you assess the parametric or non-parametric nature of some of the variables? (e.g. length of interpregnancy interval).

Response: As stated above. We have re-analyzed the distribution of all variables using the P-P plot and the skewness and kurtosis.

o Please state what statistical test was performed for each variable.

Response : Please see detailed below

Maternal characteristics / obstetrical history 

Maternal age at delivery, years, mean ±std (paired samples t-test)

Maternal BMI, median (25th, 75th) (Wilcoxon Signed Rank Test)

Parity, median (25th, 75th) (Wilcoxon Signed Rank Test)

Gestational week at delivery (paired sample t-test)

Interpregnancy interval, months, median (25th, 75th) (Wilcoxon Signed Rank Test)

Gestational diabetes N (%) Pearson's chi square test 

Hypertensive disorders N (%) Pearson's chi square test

Artificial reproductive therapy N (%) Pearson's chi square test

Multiple pregnancy N (%) Pearson's chi square test

Preterm delivery <37 weeks N (%) Pearson's chi square test

Trial of labour after CD (TOLAC) N (%) Pearson's chi square test

Vertex presentation Pearson's chi square test

Maternal complications

Uterine rupture N (%) Pearson's chi square test

Uterine scar dehiscence N (%) Pearson's chi square test

Haemoglobin drop ≥ 3g% N (%) Pearson's chi square test

Blood transfusion N (%) chi-square test 

Duration of CD, mean (minutes) std ( paired sample t-test )

Duration of CD > 90 percentile (> 62 min) N (%) Pearson's chi square test

Relaparotomy N (%) Fisher's Exact Test

Admission to the intensive care unit N (%)

Complicated CD* Pearson's chi square test

Neonatal outcome

Birth weight of newborn ± std Pearson's chi square test

Macrosomia (>4000g) N (%) Pearson's chi square test

Apgar at 5 minutes ≤ 7 N (%) Pearson's chi square test

Neonatal intensive care unit N (%) Pearson's chi square test

o Were intervariable interactions assessed before introduction to the multivariate model?

Following your comment we checked for intervariable interactions 

1. Sub-Saharan African descent and composite complicated maternal CD2

 Exp(B) 3.3 (p=0.195)

2. TOLAC at CD3 and composite complicated maternal CD2

Exp(B) 1.6 (p=0.462)

3. gestational age at CD3 and composite complicated maternal CD2

 Exp(B) 1.023 (1.012-1.033) (p<0.001)

We added two of these interactions to Table 2 

When applying the intervariable interaction of gestational age at CD3 and composite complicated maternal CD2 as a variable to the model it decreased the Nagelkerke R Square from 0.59 to 0.043

Results

• The flow chart provided by the authors greatly assists in understating the cohort composition. However, it is not clear what are the reasons for the exclusion of 1350 (50.4%) women of the 2781 women who have underwent consecutive CD2 and CD3 without placenta accreta. Please elaborate on the exclusion reasons and provide a matching table.

Thank you for bringing it to our attention that the flow chart was not clear. There were 1350 women who underwent CD2 and CD3 during the study period (2700 CDs). After excluding 38 cases of placenta accreta (19 women, 38 CDs) the study included 1331 women. 

Attached is the revised flow chart.

• Table 1:

o Please state if mean of median was used in BMI.

Response: Median was used and added to the Table. 

o Given that your institution is also caring for grandmultiparity cases, please use interquartile range when using a median (e.g. parity).

Response: The min, max, 25th and 75th quartiles were added to the variable , parity, in Table 1.

o Please state the gestational week of the pregnancies.

Response: We added gestational week to the Table and a paired t-test was employed.

o Please include all assessed variables relevant for the cohort characterization (even if non-comparable) such as ethnicity, religion and comorbidities. Especially so if those variables are later used in the multivariate model.

Response: We added to Table1, vertex presentation, origin of Sub-Saharan African descent and Non Jewish religion, 

• Page 5, line 132 – I believe that () marks were supposed to be placed around the TOLAC data.

Response: Thank you. This omission was corrected. 

• Table 2:

o Please include the p-value for each assessed variable.

Response: We divided the original Table 2 into two tables; Table 2 and Table 3.

Table 2: presenting the univariate analysis and as requested we inserted a column for the P value. For simplicity we presented the CI with one space after zero. In cases that the CI included a value of 1.0 we added a detailed description of 3 spaces after zero in the bottom of the table.

Table 3: multivariate analysis 

o Please make sure that the multivariate model includes BMI, maternal age, gestational age and parity as adjusting variables.

o Please consider reporting the final amount of cases included in the model, its p-value and -2LL.

Response: We presented the multivariate model in a new table. Including the variables suggested. 

Noteworthy- As written in the original manuscript “Given that BMI only became a required field in the medical record in the last three years, it was absent in 68% of cases of CD3. Therefore, we developed three different binomial logistic regression models.”

MODEL 1. Model with BMI 

number of cases in the model 421,

-2 LL 370.292 (initial 385.217), 

Nagelkerke R Square = 0.058

χ2(7) = 14925 (8) , p =0.061

Hosmer and Lemeshow = 0.683

MODEL 2 Model without BMI 

number of cases in the model 1328,

-2 LL 1160.835 (initial 1211.595), 

Nagelkerke R Square = 0.063

χ2(7) = 50.760, p < .001

Hosmer and Lemeshow = 0.427

o This is a suggestion only: consider establishing a second model in which the specific association of any complication in CD2 with a complication in CD3, adjusted only to BMI, maternal age, gestational age and parity. I believe that while the current model is informative, the suggested model will be easier to apply in clinic and will answer the aims of the study better.

In this context, please note that the current model has

MODEL 3 Suggested Model any complication in CD2 with a complication in CD3, adjusted only to BMI, number of cases in the model 421 ,

-2 LL 374.310 (initial 384.841), p=0.194

Nagelkerke R Square = 0.058

Hosmer and Lemeshow = 0.509

The highest Nagelkerke R Square = 0.063 was for Model 2 (without BMI) , given that BMI was documented only for 32% of women in CD3 and was not a significant factor for complication not in the univariate analysis and not in the multivariate analysis, and the model with BMI was less significant p=0.061 compared to the model without p<0.001, we preferred to adopt Model 2 which is presented in the text as Table 3. 

We added to the results the following paragraph

Line 197-204 “A binomial logistic regression was performed to determine the effect of a maternal complication in previous CD2 together with potential risk factors at CD3 on the likelihood of composite maternal complication in CD3. The model included; maternal age, parity, preterm delivery, TOLAC and birthweight of newborn, a Sub-Saharan African origin and a composite maternal complication in previous CD2. The model included 1328 cases, was statistically significant, χ2(7) = 50.760, p <0.001, explained 6.3% of the variance of composite complicated maternal CD3 and correctly classified 82.9% of cases (Table 3)”

In the discussion section we expand and present the three models Line 243-262, see below.

• Page 6, lines 143-148 – Please refrain from stating table 2 in the end of each sentence.

Response: The manuscript was revised as such. 

• While it is easy to deduce the percentage of non-complicated and complicated cases in CD2 and CD3, it is not as easy to deduce the overlap between the groups. Please provide a 2x2 table to allow for easier calculation of sensitivity, specificity, positive and negative predictive values.

We added a Table 4 to the manuscript.

Table 4: The association between a composite complicated maternal CD2 and a composite complicated maternal CD3

 composite complicated maternal CD3 

total no yes 

159 112 47 yes composite complicated maternal CD2

1172 993 179 no 

1331 1105 226 total 

Person Chi-Square < 0.001

Composite complicated maternal CD defined as either one of the following: blood transfusion > 1 packed cell, uterine rupture, dehiscence, relaparotomy or duration of operation > 90 percentile

Discussion

General notes:

• The discussion chapter of this manuscript in its current form could benefit significantly from language editing as well as self-proofing. While the general concepts the authors wish to convey are presented in this chapter, they are difficult to deduce and extract.

Response: The discussion section was revised, edited and improved with regards to readability and clarity. 

• While the reference list of this manuscript is updated and encompasses quotations from well-established journals, reference to those results in this chapter are lacking. By the end of the chapter one may still fail in understanding how do these results fit what is currently known about low-order CDs.

Furthermore, it is somewhat surprising no to find a reference to similar papers who have assessed low-order CDs in the past (e.g. PMID 16501678, 30463461)

Response: As suggested we rewrote the discussion, added 4 references and wrote the following paragraph

Line 222-233 “There are limited studies that focus solely on repeat low order CD. The majority of the studies combine low and high order repeat CD in fixed groups, and some even include the primary CD as a reference group, as such it is difficult to seek conclusions specifically on the group of low order repeat CD. [8,13,16-19]. As summarized in Table 5, most studies were retrospective, compared between low and high order CD on various outcomes, including maternal complications, focusing mainly on the effect of the order of CD [8,13,16-19]. 

This study appears to be the first longitudinal follow up study reported in the English literature of a cohort of women focusing exclusively on repeat low order CD, excluding women with placental accreta spectrum, aiming to predict maternal complications in CD3 based on specific parameters of CD2 and CD3. ”. 

Table 5: Studies evaluating maternal complications among women undergoing low order repeat CD

reference number methodology aim of the study order of CD / 

groups compared placenta accreta spectrum Maternal complications

8 Prospective observational cohort descriptive: comparing high order to low order CD CD1/ CD2 / CD3 / CD4 / CD5 / ≥ CD6 

elective CD included dehiscence / rupture / blood transfusion / bowl-bladder injury / cesarean hysterectomy/

intensive care unit admission

13 Retrospective descriptive study. descriptive: comparing high order to low order CD

 CD1/ CD2 / CD3 / ≥ CD4

elective CD included dehiscence / rupture / blood transfusion / bowl-bladder injury / wound infection/ UTI/ DVT/ cesarean hysterectomy/

duration of surgery

16 Retrospective descriptive study. descriptive: comparing high order to low order CD CD1/ CD2 / ≥ CD3

elective and emergent CD included dehiscence / rupture / blood transfusion / bowl-bladder injury / wound infection/ UTI/ DVT/ cesarean hysterectomy/ wound infection

17 Retrospective descriptive study. descriptive: comparing high order to low order CD

logistic regression model for bladder injury (2 cases) bowel injury (1 case) cesarean hysterectomy (2 cases) CD2 / ≥ CD3

elective and emergent CD included dehiscence / rupture / blood transfusion / bowl-bladder injury / wound infection/ UTI/ DVT/ cesarean hysterectomy/

duration of surgery

18 Retrospective descriptive study. descriptive: comparing high order to low order CD

 CD2 & CD3 / ≥ CD4

elective CD included dehiscence / rupture / blood transfusion / bowl-bladder injury / cesarean hysterectomy/

duration of surgery

19 Retrospective descriptive study. descriptive: comparing high order to low order CD

in regard to the timing of repeated CD after two or more previous CD CD2 / ≥ CD3

 included dehiscence / rupture / blood transfusion / bowl-bladder injury / cesarean hysterectomy/

wound complication/ DVT

• I would assume that the multivariate model in its current form will alter after the introduction of the suggested variables. Please revise the data in this segment of the manuscript as well.

Response: We rewrote the discussion in accordance to the revised multivariate analysis

• In the perspective of this reviewer, the strongest aspect of this manuscript lays in its cohort size and multivariate logistic regression model. However, this model is poorly referred to in the discussion, with specific odds ratios not being mentioned.

Response: As mentioned above, we added the following paragraph to the discussion:

Line 243-262 “Initially we developed a multivariate logistic regression model including maternal characteristics such as age, parity, gestational age, BMI, together with the significant factors identified by the univariate analysis. Given that BMI only became a required field in the medical record in the last three years of the study, it was absent in 68% of cases of CD3. Therefore, we developed three different binomial logistic regression models. The first model included BMI, age, gestational age, parity, TOLAC, birth weight, Sub-Saharan African origin and a composite complicated maternal CD2. The second model contained all the variables in the first model except for BMI. The third model, similar to the first model, but instead of a composite complicated maternal CD2, each one of the complications at CD2 was added independently to the model. Comparing the three models with regards to the number of cases included in the model, significance and Nagelkerke R Square (NRS) showed superiority to Model 2. (Model 1; n=421, p=0.061 and NRS= 0.058, Model 2; n=1328, p<0.001 and NRS= 0.063, Model 3; n=420, p=0.194 and NRS= 0.058). Of the risk factors identified in the univariate analysis, only three remained significant in the multivariate logistic regression model; composite complicated maternal CD2 aOR=2.2 p<0.001, preterm delivery aOR=1.7 p=0.010 and Sub-Saharan African origin aOR=a3.7 p<0.001. Although the model was significant and identified independent risk factors for a composite complicated maternal CD3, the area under the ROC curve was 0.639, 95% CI (0.597 0.681), indicating a poor prognostic capability of the model.”

In addition to that, this section of the manuscript requires some more specific alterations:

• Page 6 line 153 – I would suggest using a absolute numbers when stating a 40% increase as opposed to 17% of 12% (which is calculated to be 41.6%).

Response: Line 181-183 “Of the 1331 women in the cohort, 159 women (11.9%) had a composite complicated maternal CD2 compared to 226 (16.9%) of women during CD3, an increase of 67 cases (42%) p<0.001”. 

• Page 6 line 155 – it is unclear what do the authors mean by Viz.

Response: This section was removed from the current, revised manuscript

• Page 6 lines 155-156 – The use of PPV and NPV is somewhat inaccurate in the context of prediction. It can be used in reference to an already established database to evaluate the accuracy of using a test (in this case, complications at CD2) to predict an outcome (in this case, complications at CD3). It is less accepted as a predictive tool. Please amend the text accordingly.

Response: The text has been updated to reflect this change. 

Statistical analysis: Line 157-160 “The association between a composite complicated maternal CD2 and CD3 was evaluated by a chi square test. The Sn, Sp, PPV and NPV of a complicated CD2 as a diagnostic tool for predicting complications in the subsequent CD3 was calculated.”

Result section: Line 212-215 "A composite complicated maternal CD2 was found to be associated with maternal complications in the following CD3, p < 0.001. The Sn, Sp, PPV and NPV was calculated to assess the potential use of a "composite complicated maternal CD" as a "diagnostic tool" for maternal complications in the following CD3; 21%, 90%, 30% and 85% respectively. Table 4."

Discussion: Line 234-242 “…a composite complicated maternal CD2 was associated with maternal complications in CD3, p < 0.001. (Table 4) However, the use this association as an independent "diagnostic tool" for a complicated CD3 in future studies is weak given that the Sn and PPV are low. Of the 226 cases with a composite complicated maternal CD3, only 47 (21%) experienced a complication in the previous CD Sn=21% and of the 159 women with a composite complicated maternal CD2, 112 (70%) were not complicated in CD3, PPV=30%. Attempting to use this association as a "diagnostic tool" in practice is weak given that the Sn and PPV are low. Of the 226 cases with a composite complicated maternal CD3 only 47 (21%) experienced a complication in previous CD2 Sn=21% and of the 159 women with a composite complicated maternal CD2, 112 (70%) were not complicated in the following CD3, PPV=30%.“

• Page 7 lines 173-175 – if I understand correctly, this group had a threefold risk for maternal complicated CD3. Please correct the text if this is correct.

We corrected the text and wrote: 

Line 271-275 “A small percentage of the study population (n=39, 3%,) were of Sub-Saharan African descent (Ethiopian), who had an OR of 3.7 for a composite complicated maternal CD3 compared to Caucasian women. This is, to some extent, similar to the association found between the presence of keloids/dense pelvic adhesions in African American women [23]”. 

• Strengths – It is suggested that the authors include limited variability in surgical technique, as expected of a one center study, as another strength of this study.

Response: We added to the strengths:

Line 322-323 “(5) Given that the study was performed in one clinical center we expect limited variability significance in surgical technique.”

Conclusion

• The discussion barely refers to the difference in surgical duration and even tat only at the limitations segment. Therefore, it is not clear why the authors have decided to include the time of previous caesarean delivery as one of the important factors when planning a repeat CD. Either elaborate further on this finding, how it refers to currently existing knowledge and literature and why should it be interpreted the way you think it does, or omit it from the conclusion please.

Response: We elaborated on the importance of operative time as a practical risk marker for maternal complication as we added to the discussion section. In addition we changed the conclusion section 

Line 288-297 " There is a strong association between length of operation and presence of adhesions, adhesiolysis and maternal complications [11-13]. In the current study previous prolonged operative time (> 90th percentile at CD2) increased risk for a maternal complication in CD3 by an OR of 2.4, p<0.001. When applying each one of the complications of CD2 independently, instead of as a composite, (logistical regression Model 3), the predictor variable of prolonged operative time at CD2 remained statistically significant (aOR of 2.6, p=0.013). As such, operative time is an important risk factor for a maternal complication in the subsequent CD. Since operative time is a mandatory field in most electronic medical records and its accessibility make it a practical and important factor available for use when counseling women about the risk of complications in future CDs." 

Reviewer #2: 

- Manuscript should be further revised by a native English speaker.

Response: A professional, native English speaker language editor has reviewed and revised the manuscript.

- Does this manuscript conform the Enhancing the QUAlity and Transparency Of health Research (EQUATOR) network guidelines? It would be mandatory to declare about this element.

Response: We added the following sentence to the method section :

Line 170-171 “The manuscript is presented according to the STROBE guidelines [15]”.

- Recent and novel evidence suggested that epigenetic changes, in particular altered expression of selective miRNA, may play a key role in both placental-induced diseases such intrauterine growth restriction. It would be mandatory to discuss (at least briefly) this topic, referring to: PMID: 28466013; PMID: 20104830.

Response: We thank the reviewer and added to the discussion the following paragraph

Line 276-281 “There is increasing data regarding the role of microRNAs in cell development, differentiation, and proliferation. This family of small noncoding RNAs, ranging 22 nucleotides in length, regulates gene expression. Their role in many human diseases has been demonstrated including adhesion formation, preeclampsia and intrauterine growth retardation [24, 25]. Genetic variations between patients and ethnic origin could be attributed in part to variations in micro RNAs.”

- The real challenge in the era of molecular medicine is to find a biomarker, or even better a panel of biomarkers, for early diagnosis of pre-eclampsia, intrauterine growth restriction and stillbirth. I would stress this point, referring to: PMID: 28243732; PMID: 35245721.

Given that the main focus of the study is on maternal complications following repeat CD We combined to the reviewers comment also the aspect of biomarkers and sonographic findings to adhesions. 

We added to the discussion: 

Line 281-287 “The search for an association between biomarkers or sonographic findings with specific diseases or pathological conditions including preeclampsia, stillbirth and IUGR is underway [26, 27]. Studies have shown an association between specific biomarkers and severity of adhesions in endometriosis and sonographic findings associated with the severity of adhesions in repeat CDs [28, 29]. Future studies focusing on a complication of repeat CD could combine these factors with the clinical risk factors identified in the current study.”________________________________________

I wish to thank you again for the opportunity to improve and resubmit this manuscript. I sincerely hope that I have sufficiently addressed all the reviewers’ comments and concerns and that it is deemed worthy for publication in PLOS ONE.

Respectfully submitted, 

Orna Reichman

---

## [Decision Letter · Decision Letter 1]

17 Oct 2022

Repeat low order caesarean delivery, risk factors for complications; a retrospective longitudinal study.

PONE-D-22-18931R1

Dear Dr. Reichman,

We’re pleased to inform you that your manuscript has been judged scientifically suitable for publication and will be formally accepted for publication once it meets all outstanding technical requirements.

Kind regards,

Antonio Simone Laganà, M.D., Ph.D.

Academic Editor

PLOS ONE

Additional Editor Comments (optional):

Authors performed the required corrections, which were positively evaluated by the reviewers. I am pleased to accept this paper for publication.

Reviewers' comments:

Reviewer's Responses to Questions

**Comments to the Author**

1. If the authors have adequately addressed your comments raised in a previous round of review and you feel that this manuscript is now acceptable for publication, you may indicate that here to bypass the “Comments to the Author” section, enter your conflict of interest statement in the “Confidential to Editor” section, and submit your "Accept" recommendation.

Reviewer #2: (No Response)

2. Is the manuscript technically sound, and do the data support the conclusions?

Reviewer #2: Yes

3. Has the statistical analysis been performed appropriately and rigorously? 

Reviewer #2: Yes

4. Have the authors made all data underlying the findings in their manuscript fully available?

Reviewer #2: Yes

5. Is the manuscript presented in an intelligible fashion and written in standard English?

Reviewer #2: Yes

6. Review Comments to the Author

Reviewer #2: I read with great interest the Manuscript titled “Repeat low order caesarean delivery, risk factors for complications; a retrospective longitudinal study” (PONE-D-22-18931R1), which falls within the aim of this Journal. In my honest opinion, the topic is interesting enough to attract the readers’ attention. Methodology is accurate and conclusions are supported by the data analysis.

7. PLOS authors have the option to publish the peer review history of their article (what does this mean?). If published, this will include your full peer review and any attached files.

Reviewer #2: **Yes: **Pietro Serra

---

## [Editor Report · Acceptance letter]

24 Oct 2022

PONE-D-22-18931R1 

Repeat low order caesarean delivery, risk factors for complications: A retrospective, longitudinal study. 

Dear Dr. Reichman:

I'm pleased to inform you that your manuscript has been deemed suitable for publication in PLOS ONE. Congratulations! Your manuscript is now with our production department. 

Kind regards, 

on behalf of

Dr. Antonio Simone Laganà 

Academic Editor

PLOS ONE